# Two HEPN domains dictate CRISPR RNA maturation and target cleavage in Cas13d

Bo Zhang [1,2,3,6], Yangmiao Ye[1,6], Weiwei Ye[1], Vanja Perčulija[1], Han Jiang[1], Yiyang Chen[1], Yu Li[1], Jing Chen[1], Jinying Lin[1], Siqi Wang[1], Qi Chen[1], Yu-San Han[4] & Songying Ouyang [1,2,3,5]

Cas13d, the type VI-D CRISPR-Cas effector, is an RNA-guided ribonuclease that has been repurposed to edit RNA in a programmable manner. Here we report the detailed structural and functional analysis of the uncultured *Ruminococcus sp.* Cas13d (UrCas13d)-crRNA complex. Two hydrated $Mg^{2+}$ ions aid in stabilizing the conformation of the crRNA repeat region. Sequestration of divalent metal ions does not alter pre-crRNA processing, but abolishes target cleavage by UrCas13d. Notably, the pre-crRNA processing is executed by the HEPN-2 domain. Furthermore, both the structure and sequence of the nucleotides U(-8)-C(-1) within the repeat region are indispensable for target cleavage, and are specifically recognized by UrCas13d. Moreover, correct base pairings within two separate spacer regions (an internal and a 3'-end region) are essential for target cleavage. These findings provide a framework for the development of Cas13d into a tool for a wide range of applications.

[1] The Key Laboratory of Innate Immune Biology of Fujian Province, Biomedical Research Center of South China, Key Laboratory of OptoElectronic Science and Technology for Medicine of Ministry of Education, College of Life Sciences, Fujian Normal University, 350117 Fuzhou, China. [2] Provincial University Key Laboratory of Cellular Stress Response and Metabolic Regulation, College of Life Sciences, Fujian Normal University, 350117 Fuzhou, China. [3] Laboratory for Marine Biology and Biotechnology, Pilot National Laboratory for Marine Science and Technology (Qingdao), 266237 Qingdao, China. [4] Institute of Fisheries Science, College of Life Science, National Taiwan University, 1, Sec. 4, Roosevelt Rd., 10617 Taipei, Taiwan. [5] National Laboratory of Biomacromolecules, Institute of Biophysics, Chinese Academy of Sciences, 100101 Beijing, China. [6] These authors contributed equally: Bo Zhang, Yangmiao Ye. Correspondence and requests for materials should be addressed to S.O. (email: ouyangsy@fjnu.edu.cn)

                                                                          1

CRISPR-Cas system is a heritable, adaptive immune system that protects bacteria and archaea against phages and other invasive MGEs[1–4]. The most prominent feature of the system is the array of genomic DNA known as clustered regularly interspaced short palindromic repeats (CRISPR). CRISPR serves as an archive that stores information about previous infections in form of DNA sequences (termed spacers) acquired from the invading MGEs[5–7]. CRISPR-Cas systems can be assigned to class 1 and class 2 systems, which are further divided into types and subtypes[2,4,6]. Class 2 systems (including types II, V, and VI) deploy a single Cas effector protein to accomplish CRISPR RNA (crRNA) biogenesis and interference[4,6,8]. Due to their simplicity, class 2 systems have become powerful tools for genome editing and other applications in life sciences[9–16].

Particularly interesting among the class 2 systems is the recently identified type VI system, which encompasses subtypes VI-A (effector Cas13a, also known as C2c2), VI-B (effector Cas13b), VI-C (effector Cas13c), and VI-D (effector Cas13d)[14,17–21]. Cas13 family is the only family of class 2 Cas enzymes known to exclusively target single-stranded RNA. RNA cleavage is mediated by two $R-X_4-H$ motifs, which are characteristic of higher eukaryotes and prokaryotes nucleotide (HEPN)-binding domains[22–26]. Typically, Cas13 effectors exhibit collateral non-specific RNase activity triggered upon binding target RNA sequence[17,19–21,26]. Also, Cas13 enzymes have been engineered as programmable RNA-binding modules for targeting and editing RNA[19,20,27–30].

Analogously to the other members of the Cas13 superfamily, Cas13d also contains two $R-X_4-H$ HEPN motifs, but bears little overall similarity to the amino acid sequences of Cas13a and Cas13b[17–21]. Owing to its biochemical properties, especially small size, lack of constraints on the target flanking sequences, and highly efficient and specific target RNA cleavage, Cas13d is a promising candidate for RNA engineering and editing[20,21]. However, more widespread application is hindered by the lacking structural analysis of mechanisms underlying Cas13d activity.

Recently, cryo-EM structures of *Eubacterium siraeum* Cas13d (EsCas13d) in crRNA-bound, and target-bound states have been reported[31]. Nevertheless, some important issues remain unanswered or require further validation. For example, the pre-crRNA processing site has not been identified. Moreover, an electron density that likely represents a $Mg^{2+}$ ion was found to interact with the highly ordered nucleotides at the 3′-end of the crRNA repeat region both in EsCas13d binary and ternary complexes[31]. Due to limited resolutions of the cryo-EM structures[31], this prominent feature should be inspected in a higher-resolution structure of Cas13d. Besides, two different results were reported regarding the dependence of pre-crRNA processing on $Mg^{2+}$ ion[21,31], which requires further verification. Whether the crRNA repeat region is recognized in both structure-specific and sequence-specific manner needs detailed investigation. Last but not least, the mismatch tolerance between the crRNA spacer region and the target RNA should be thoroughly tested. Here, we try to answer these questions by using high-resolution crystal structures of Cas13d and biochemical studies, and provide additional molecular information that can be used in rational design of the CRISPR-Cas13d system for a wide range of potential applications.

## Results

### Overall structure of the UrCas13d-crRNA binary complex.
In order to understand the structural basis of pre-crRNA processing and mature crRNA recognition by Cas13d, we solved the crystal structure of the uncultured *Ruminococcus sp.* Cas13d (UrCas13d,

also known as RspCas13d)-crRNA complex at 2.15 Å resolution using the single-wavelength anomalous diffraction method (Supplementary Table 1). The binary complex was obtained by the selenomethionine (SeMet)-derived UrCas13d (R288A/R823A) mutant co-expressed with CRISPR RNA in vivo. The SeMet-labeled structure was subsequently used as a model to solve the crystal structure of the unlabeled UrCas13d (R288A/R823A) binary complex at 1.86 Å resolution using the molecular replacement method.

Due to its minimal size among type VI CRISPR effectors, UrCas13d exhibits an atypically compact architecture (Fig. 1a–c and Supplementary Movie 1). Although the conventional bilobed architecture consisting of a recognition (REC) lobe and a nuclease (NUC) lobe is still discernible in UrCas13d, its compact structure blurs this bilobed notion to certain extent. The REC lobe consists of the N-terminal domain (NTD) and Helical-1 domain (Fig. 1a–d). The NUC lobe consists of two HEPN domains (HEPN-1 and HEPN-2) and Helical-2 domain. The mature crRNA is sandwiched within a positively charged channel formed by the REC and NUC lobes (Fig. 1d–h).

In the binary complex, except for NTD, the predominant secondary structures of UrCas13d are α-helices (Supplementary Fig. 1). NTD contains a seven-stranded, antiparallel β-sheet and short α-helix. HEPN-1 is composed of ten α-helices, and HEPN-2 domain is composed of eleven α-helices. Two HEPN domains interact with each other primarily through helix-α3 and helix-α28. Notably, the long helix-α28 of HEPN-2 domain extends to the opposite sides of the binary complex and roughly runs parallel to the crRNA structure. Helical-1 domain contains eight α-helices. Helical-2 domain contains eight α-helices, and wraps around helix-α28 of HEPN-2 domain.

A mature 50-nt crRNA was determined in the binary complex (Fig. 1b, e and Supplementary Fig. 2). It originated from an in vivo transcript encoded by a co-expressed CRISPR template, which was further processed by UrCas13d itself in *E. coli* cells. The crRNA consists of a 30-nt repeat region (C(-30)-C(-1)) and 20-nt spacer region (U(1)-A(20)) (Fig. 1e). The repeat region displays a stem-loop architecture. Its secondary structure is remarkably similar to the predicted one, with the exception of base pairing of A(-29) (Fig. 1b and Supplementary Fig. 3). The stem region shows a distorted A-form RNA duplex, which includes seven Watson-Crick base pairs and a wobble G-U base pair (Fig. 1b). The loop region contains four nucleotides (A(-20)-U(-17)). Interestingly, the nucleotides U(-8)-C(15) display four successive U-shaped turns (Fig. 1e).

### Hydrated $Mg^{2+}$ ions aid in stabilizing the crRNA repeat region.
The high resolution of the binary complex unambiguously shows an octahedral-shaped electron density around the phosphate group of the nucleotide C(-7). Considering that $Mg^{2+}$ and $Na^+$ are metal ions provided in buffer systems during the purification and crystallization, a penta-hydrated $Mg^{2+}$ was built into the present structure. The magnesium ion directly coordinates with five water molecules and an oxygen atom from the phosphate group of the nucleotide C(-7) (Fig. 2a). Moreover, the $(Mg(H_2O)_5)^{2+}$ ion forms hydrogen bonds with the phosphate groups of the nucleotides A(-5), A(-3), and A(-2), as well as the side chain of residue D178 (Fig. 2a, b and Supplementary Movie 2). The detailed interactions between the $(Mg(H_2O)_5)^{2+}$ ion and the surrounding chemical groups are shown in Fig. 3. In terms of position, the hydrated $Mg^{2+}$ is situated in the center of an U-shaped turn formed by the nucleotides U(-8)-A(-2) of the repeat region. Notably, residues D178, K181 and K524 are involved in stabilizing the conformation of the $(Mg(H_2O)_5)^{2+}$ ion and the coordinated nucleotides (Fig. 2b). These residues are

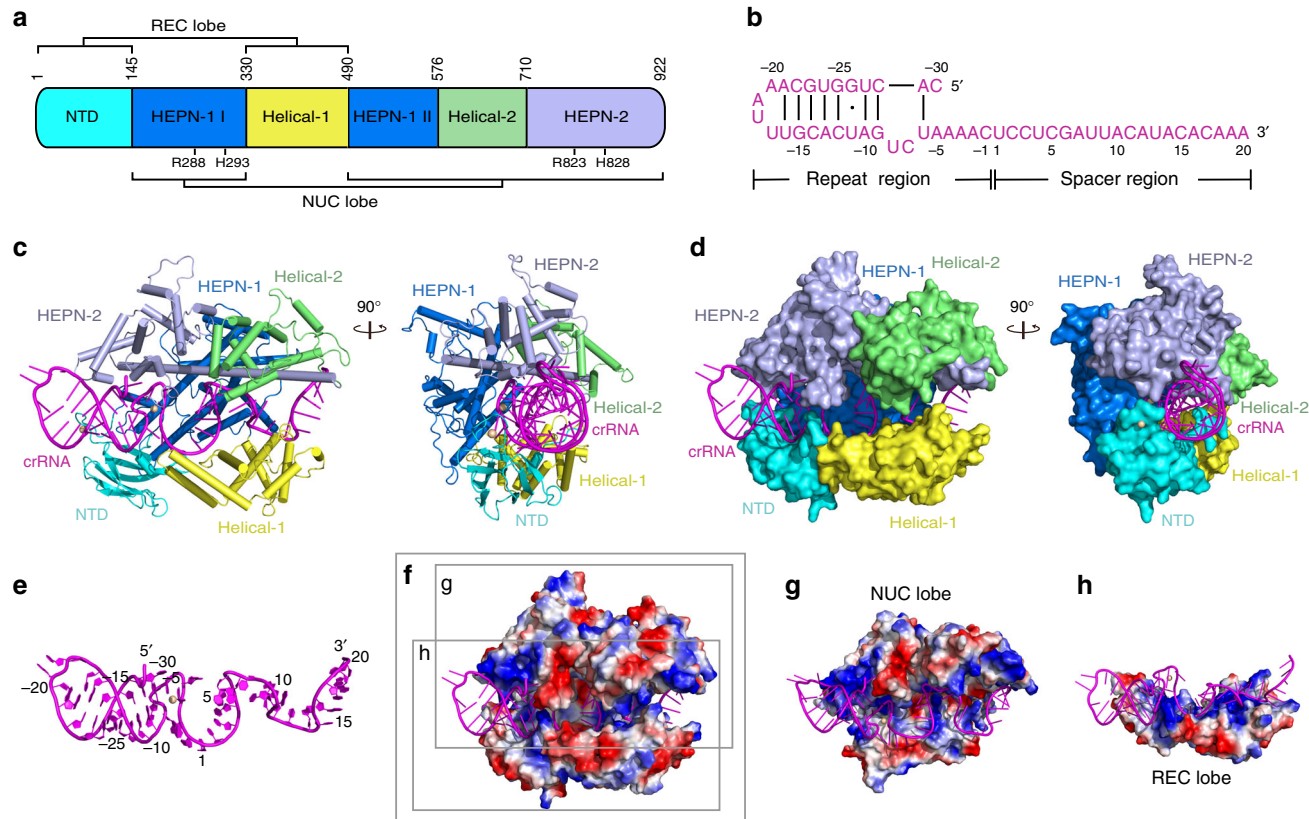

**Fig. 1** Overall structure of the UrCas13d-crRNA binary complex. **a** Domain organization of UrCas13d. Catalytic residues of the R-X$_4$-H motifs within two HEPN domains are labeled. **b** Schematic representation of the crRNA secondary structure. Bars between nucleotide pairs represent Watson-Crick base pairs and dot represents wobble G-U base pair. **c** Overall structure of the UrCas13d-crRNA binary complex shown in two different orientations, color-coded as defined in **a** and **b**. **d** Surface representations of the UrCas13d-crRNA binary complex shown the same views as in (**c**). **e** Structure of crRNA in the binary complex. Mg$^{2+}$ is colored in wheat. **f** Electrostatic potential surface of the crRNA binding channel in the binary complex. Red, white and blue indicate negative, neutral and positive electrostatic potential surfaces, respectively. **g** Electrostatic potential surface of the NUC lobe in complex with crRNA. **h** Electrostatic potential surface of the REC lobe in complex with crRNA

conserved among Cas13d family members (Supplementary Fig. 4), and substituting any of them significantly decreases target RNA cleavage (Fig. 2c). Therefore, the penta-hydrated Mg$^{2+}$ ion plays an important role in stabilizing the conformation of the crRNA repeat region.

In addition, a tetra-hydrated Mg$^{2+}$ was found to coordinate with two backbone oxygen atoms of residues R137 and E140 from a loop region of NTD (Fig. 2d, e and Supplementary Movie 3). This (Mg(H$_2$O)$_4$)$^{2+}$ ion forms hydrogen bonds with backbone atoms of residues T135, R137, S141, and S142, which helps to maintain the conformation of this loop region. Meanwhile, the side chains of H136 and S138 point outwards and form hydrogen bonds with the phosphate groups of U(-24) and G(-25), respectively. The H136A mutation slightly decreases target RNA cleavage (Fig. 2c). These imply that the (Mg(H$_2$O)$_4$)$^{2+}$ ion is involved in the conformational stabilization of the crRNA repeat region.

To investigate whether metal ions are essential for pre-crRNA processing and target cleavage by UrCas13d, we carried out the cleavage assays with different types of metal ions. A pre-crRNA containing a repeat-spacer-repeat sequence was designed to serve as the processing target. The results clearly show that all tested divalent metal ions including Mg$^{2+}$ are unnecessary for pre-crRNA processing, but essential for target RNA cleavage (Fig. 2f, g).

**Specific recognition of the crRNA repeat region by UrCas13d.** Conformationally, the U-shaped turn of the crRNA repeat region

is stabilized not only by the penta-hydrated Mg$^{2+}$, but also by extensive interactions with residues of UrCas13d (Figs. 3 and 4a). Although no residue directly obstructs base pairing between A (-29) and U(-8), the nucleotide A(-29) pairs with U(-6) (Fig. 1b). U(-8) and C(-7) swing away from the stem duplex and form a bubble in the repeat region. The (Mg(H$_2$O)$_5$)$^{2+}$ ion coordinates with the phosphate group of C(-7), which apparently affects the conformational arrangement of U(-8) and C(-7). Besides, U(-8) stacks with A(-2), residue K56 interacts with the phosphate group of U(-8), and the side chain of Q171 forms hydrogen bond with the 2′-hydroxyl group of U(-8). The pyrimidine ring of C(-7) forms hydrogen bond with the side chain of N167, and also stacks with the side chains of Q171 and R145 (Fig. 4a, b). In addition, the side chain of R145 and the backbone nitrogen atom of M148 interact with the phosphate group of U(-6). All interactions described above help with the formation of the A(-29):U(-6) pairing. It should be noted that, when compared with a standard A-form RNA, nucleotides U(-8), A(-4), and A(-3) within the U-shaped turn have unusual conformations with respect to their sugar pucker or glycosyl bond (Supplementary Fig. 5).

Although the nucleotides A(-5)-C(-1) of the repeat region are unpaired, they are recognized in both the sugar-phosphate backbone-dependent and base-specific manners (Fig. 4a, b). The nucleotide A(-5) stacks with U(-6) and the side chain of N906 hydrogen bonds with the 2′-hydroxyl group of A(-5) (Fig. 4a). Residues N900 and R903 interact with the phosphate group of A(-5). The purine ring of A(-4) is flipped outwards to

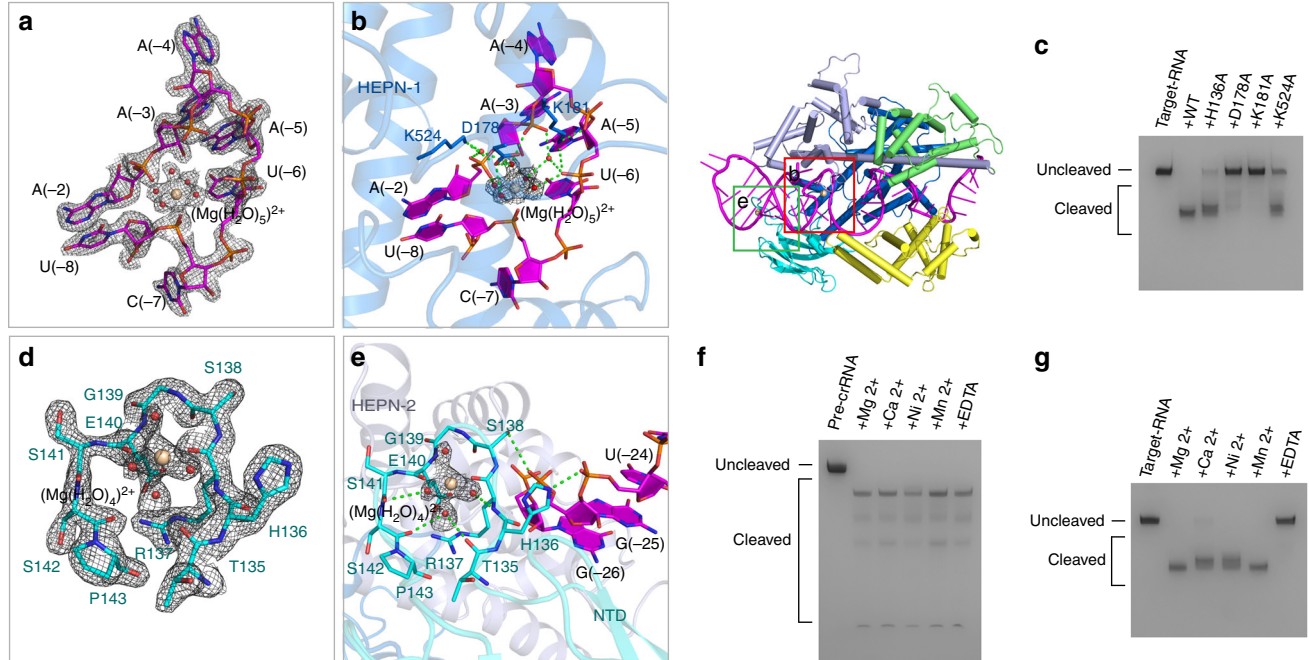

**Fig. 2** Hydrated $Mg^{2+}$ ions aid in stabilizing the crRNA repeat region. **a** The electron density map of the penta-hydrated $Mg^{2+}$ ion and its surrounding nucleotides in the crRNA repeat region. The 2Fo-Fc omit map was contoured at 1.0 σ level. **b** Details of the interactions between the penta-hydrated $Mg^{2+}$ ion and its surrounding nucleotides, residues in the binary complex. Hydrogen bonds are shown as dashed lines. Domains, residues and crRNA are colored according to Fig. 1. **c** Denaturing gel demonstrating the cleavage of target RNA by wild-type UrCas13d and the mutants in complex with crRNA. **d** The electron density map of the tetra-hydrated $Mg^{2+}$ ion and its surrounding residues of UrCas13d. The 2Fo-Fc omit map was contoured at 1.0 σ level. **e** Details of the interactions between the tetra-hydrated $Mg^{2+}$ ion and its surrounding residues, nucleotides in the binary complex. **f** Denaturing gel demonstrating pre-crRNA cleavage by wild-type UrCas13d in the presence of different divalent metal ions or EDTA. **g** Denaturing gel demonstrating the cleavage of target RNA by wild-type UrCas13d in complex with crRNA and in the presence of different divalent metal ions or EDTA

form hydrogen bond with the side chain of D914, and stacks with the side chains of R744 and R915. The side chains of K181 and R903 interact with the phosphate group of A(-4), and the side chain of R915 interacts with the 2′-hydroxyl group of A(-4) (Fig. 4a). The side chain of K181 also interacts with the phosphate group of A(-3). Furthermore, the purine ring of A(-2) is recognized by the side chains of K56, K61, N79 and N527 (Fig. 4b). The pyrimidine ring of C(-1) is recognized by G78 and R431, and partially stacks with the side chain of F75. The 2′-hydroxyl group of C(-1) hydrogen bonds with the side chain of K52.

To understand the functional importance of residues involved in recognition of the repeat region, we performed mutational analysis. As shown in Supplementary Fig. 6, single mutations of all aforementioned residues did not influence pre-crRNA processing to a detectable level. In contrast, single mutations D178A, K181A, and K524A mentioned above, as well as single mutations K61A, R744A, and R903A, greatly decrease target RNA cleavage. In addition, single mutations K56A, R145A, and Q171A also decrease target RNA cleavage to a certain level (Fig. 4c).

**Rational engineering of the crRNA repeat region.** The crRNA extends over the long axis of protein and interacts with every domain of UrCas13d (Fig. 1c, d). A large number of positively charged residues interact with the phosphodiester backbone of crRNA (Fig. 3). Notably, the nucleotides U(-24)-A(-13) of the stem-loop structure protrude from UrCas13d and are mostly exposed to the environmental solvent (Fig. 4d). Deletion of four Watson-Crick base pairs (U(-24)-A(-21)/U(-16)-A(-13)) within this exposed region (designated the Δstem mutant) has no obvious effect on pre-crRNA processing and target RNA cleavage (Fig. 4e, f). Moreover, the time-course study shows that, when

compared with pre-crRNA or crRNA, their respective Δstem mutants have a similar efficiency in pre-crRNA processing and target RNA cleavage by UrCas13d and its mutants (Supplementary Fig. 7). The rest of the stem duplex is bound inside the channel formed by NTD, HEPN-1, and HEPN-2 domains. The sugar-phosphate backbone of this bound duplex interacts with the side chains of S53, S54, S138, R145, K799, N800, and K905 (Figs. 2e, 3 and 4b). NTD resides on the stem duplex and contributes most interactions with the crRNA repeat region (Fig. 3).

**Importance of the crRNA repeat region for target RNA cleavage.** To investigate the functional importance of the nucleotides U(-8)-C(-1) from the repeat region, we carried out various substitutions and deletions in this region and further inspected their pre-crRNA processing and target RNA cleavage capabilities. The results show that single deletions of A(-4), C(-7), and U(-8) decrease pre-crRNA cleavage, and single mutations A(-2)G, A(-4)U, A(-4)G, and C(-7)G have the similar effect (Fig. 4g and Supplementary Fig. 8a). Notably, single mutation or deletion of A(-2), A(-4), C(-7), or U(-8) significantly compromises target RNA cleavage (Fig. 4h). Therefore, both structure and sequence of the repeat region are critical for target RNA cleavage.

**Recognition of the spacer region by UrCas13d.** The spacer region of crRNA forms three U-shaped turns and is anchored inside the channel formed by all domains of UrCas13d except NTD (Fig. 1c–e). The recognition of the spacer region by UrCas13d is mostly dependent on its sugar-phosphate backbone (Fig. 3). Helical-1 domain contributes multiple interactions with the spacer region. More precisely, residues K370, G373, K439, and Y443 interact with the nucleotides U(1)-U(4) (Fig. 5a), whereas residues N337, K367, S398, K402, and K405 interact with

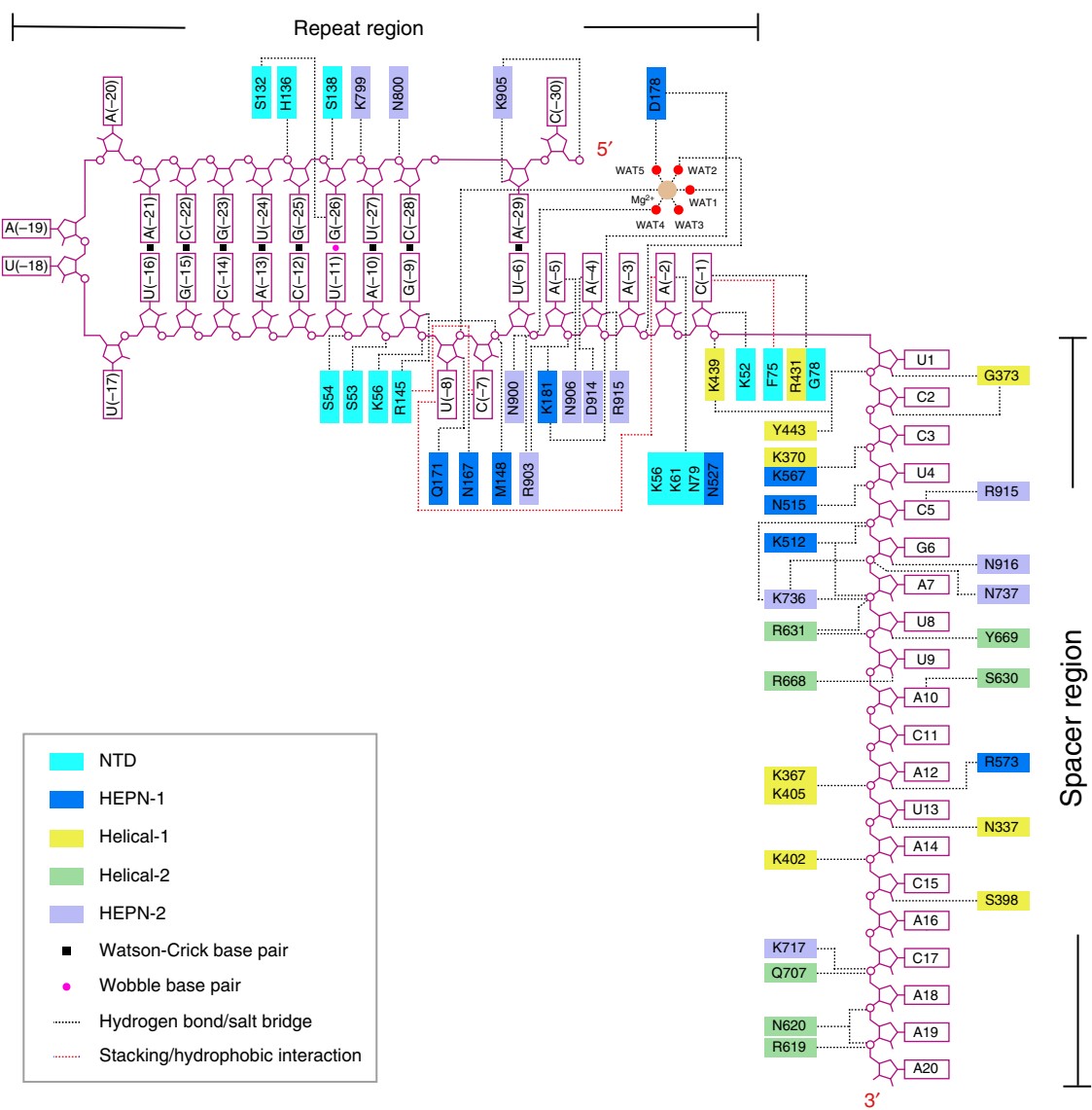

**Fig. 3** Schematic of crRNA recognition by UrCas13d. Domains and residues are colored according to Fig. 1. Hydrogen bonds and salt bridges are shown as black dashed lines. Stackings and hydrophobic interactions are shown as red dashed lines. The coordinated water molecules of the $Mg^{2+}$ ion are labeled as WATs

the nucleotides U(13)-C(15) (Fig. 5b). Moreover, the nucleotides C(5)-A(12) are recognized by residues from Helical-2 and two HEPN domains (Fig. 5b, c). Residue K512 interacts with the phosphate groups of G(6), while residue K736 simultaneously interacts with the phosphate groups of G(6), A(7), and U(8). The side chain of R631 stacks with the purine ring of A(10) and interacts with the phosphate groups of U(9) and U(8). In addition, the 3'-tail of the nucleotides A(16)-A(20) extends to a positively charged groove formed by Helical-2 and HEPN-1 domains. The phosphate backbone of this region is stabilized mainly by residues from Helical-2 domain (Fig. 5d).

To further explore the functional importance of residues involved in interactions with the spacer region, the mutational studies were performed. The results indicate that single mutations N515A, N620A, and K736A unambiguously decrease target RNA cleavage (Fig. 5e, f).

**Pre-crRNA processing site within HEPN-2 domain**. In line with the previous studies[20,21], the mature crRNA containing a 30-nt repeat region was determined in the high-resolution UrCas13d

binary complexes (Fig. 1b, e). The electron density of the nucleotides C(-30)-C(-1) is clear (Supplementary Fig. 2). To understand how pre-crRNA was processed by UrCas13d, residues around the nucleotide C(-30) were selected to carry out the mutational studies (Fig. 6a and Supplementary Movie 4). The L803A, C806A, K887A, K891A, F904A, I909A, and Q911A mutations were found to have little effect on pre-crRNA cleavage (Fig. 6b). Notably, the K905A mutation abolishes pre-crRNA cleavage. Besides, the R802A mutation significantly decreases this activity (Fig. 6b). Protein sequence alignment shows that these two residues are conserved within the Cas13d family members (Supplementary Fig. 4). Thus, the mutational and structural analyses indicate that residues R802 and K905 from HEPN-2 domain are critical for pre-crRNA cleavage, which possibly adopts a base-catalyzed mechanism.

**Active site for target RNA cleavage within two HEPN domains**. The previous study has found that two R-$X_4$-H HEPN ribonuclease motifs of Cas13d are responsible for target RNA cleavage and that the mutation of four catalytic residues in R-$X_4$-H motifs

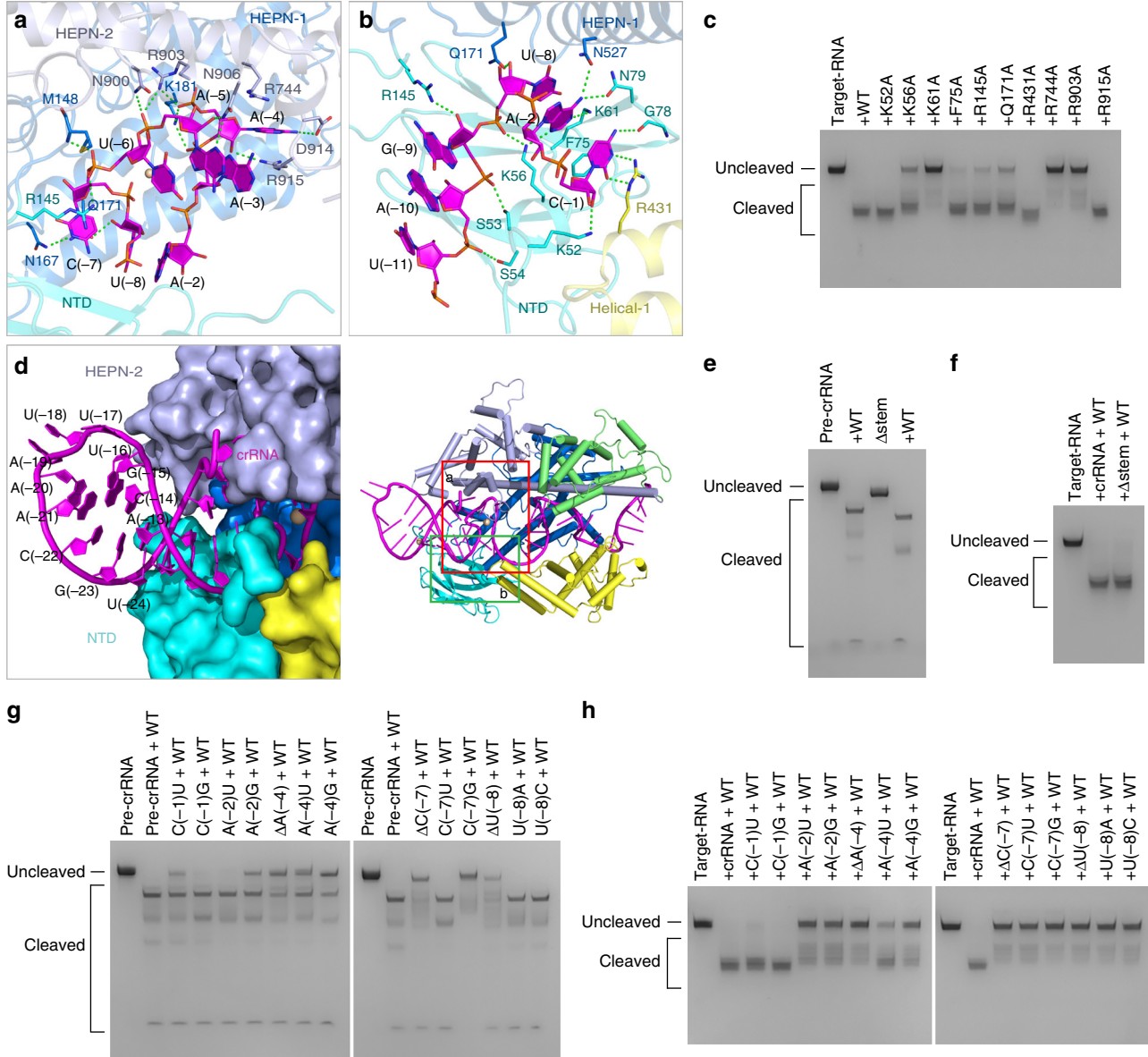

**Fig. 4** Structure-specific and sequence-specific recognition of the crRNA repeat region. **a** and **b** Details of the interactions between the crRNA repeat region and UrCas13d. **c** Denaturing gel demonstrating the cleavage of target RNA by wild-type UrCas13d and the mutants in complex with crRNA. **d** The crRNA repeat region is bound inside the channel formed by NTD, HEPN-1, and HEPN-2 domains of UrCas13d. The nucleotides U(-24)-A(-13) of the crRNA repeat region protrude from UrCas13d. **e** Denaturing gel demonstrating the cleavage of pre-crRNA or the Δstem mutant (the deletion of the nucleotides U(-24)-A(-21)/U(-16)-A(-13) in pre-crRNA) by wild-type UrCas13d. **f** Denaturing gel demonstrating the cleavage of target RNA by wild-type UrCas13d in complex with crRNA or the Δstem mutant (the deletion of the nucleotides U(-24)-A(-21)/U(-16)-A(-13) in crRNA). **g** Denaturing gel demonstrating the cleavage of pre-crRNA or the pre-crRNA mutants by wild-type UrCas13d. **h** Denaturing gel demonstrating the cleavage of target RNA by wild-type UrCas13d in complex with crRNA or the crRNA mutants

abolishes the cleavage activity[20]. We further observed that a single R288A or H828A mutation of UrCas13d was sufficient to abrogate target RNA cleavage, the H293A mutant had a greatly decreased cleavage activity, whereas the R823A mutant still showed detectable activity to a certain degree (Fig. 5f). In the present model, the distance between the backbone Cα atoms of residues R288 and R823 of UrCas13d is ~15.3 Å (Fig. 6c and Supplementary Movie 5). The corresponding distances in LshCas13a and BzCas13b are ~17.8 Å and ~18.6 Å, respectively (Fig. 6d, e). This implies that, in comparison with LshCas13a and BzCas13b, UrCas13d requires a rather small conformational rearrangement to achieve its activation, which is consistent with the compact architecture of Cas13d.

**Mismatch tolerance between the spacer region and target RNA.** PAM-adjacent seed region within the crRNA spacer is known to be essential for target recognition in Cas9, Cas12a, and Cas12b[32–36]. Unlike these Cas effectors, Cas13a possesses a central seed region[17,26,37]. In the case of the UrCas13d binary complex, both internal nucleotides U(4)-U(8) and 3′-end nucleotides A(14)-A(20) within the spacer region are exposed and accessible to the environmental solvent. This may allow UrCas13d to use the internal or 3′-end nucleotides of the spacer to interrogate target RNA. To identify the mismatch tolerance between the spacer and target RNA, we prepared 11 target RNA variants by tiling 2-nt mismatches across the 22-nt targeted region and performed the cleavage assays (Fig. 6f and Supplementary Fig. 8b). The results

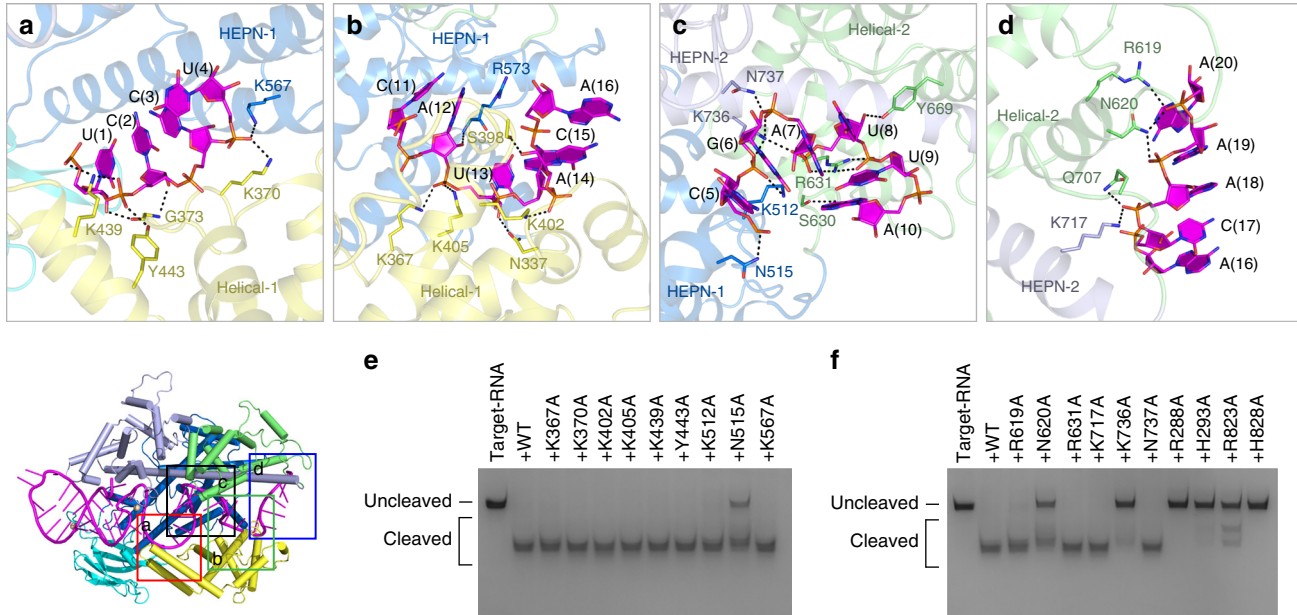

**Fig. 5** Recognition of the crRNA spacer region by UrCas13d. **a–d** Details of the interactions between the crRNA spacer region and UrCas13d. Hydrogen bonds are shown as dashed lines. Domains and residues are colored according to Fig. 1. **e, f** Denaturing gels demonstrating the cleavage of target RNA by wild-type UrCas13d and the mutants in complex with crRNA

demonstrated that mismatches of the nucleotides C(5)-U(8) and U(13)-C(22) abolished target cleavage (Fig. 6g). This suggests that correct base pairings within two separate spacer regions (an internal and a 3′-end region) are essential for target RNA cleavage.

## Discussion

To further investigate the structural differences between the UrCas13d-crRNA complex and the EsCas13d-crRNA complex, the protein backbones of two binary complexes were superimposed. The overall architectures of both binary complexes are compact and similar, with the root-mean-square deviation (RMSD) of 1.62 Å between 735 aligned Cα atoms of UrCas13d (922 amino acid residues) and EsCas13d (954 amino acid residues) (Supplementary Fig. 9a). Relative to the UrCas13d binary complex, the exposed stem-loop structure of the crRNA repeat region undergoes a ~3 Å movement towards the solvent environment in the EsCas13d binary complex. In the respective complexes, there are obvious differences in the exact position and orientation of the 5′-end nucleotide C(-30) (Supplementary Fig. 9b). C(-30) in the UrCas13d binary complex is located between two critical residues (R802 and K905) and the C-terminus of UrCas13d. Such positioning of C(-30) may be favorable for the pre-crRNA processing. In contrast, C(-30) in the EsCas13d binary complex swings away from two critical residues (R828 and K932) of pre-crRNA processing site, and the C-terminus of EsCas13d is located between these two residues and C(-30). In addition, in the target RNA-cleavage sites of two binary complexes, two catalytic residues (H293 of UrCas13d and H300 of EsCas13d) are reversely located in the respective HEPN-1 domains (Supplementary Fig. 9c).

Current availability of the type VI CRISPR-effector structures allows us to further analyze structural and functional features of Cas13d by comparing similarities and differences between Cas13a, Cas13b, and Cas13d[25,26,31,38,39]. Overall, the domain architecture of Cas13d shares certain spatial consistency with Cas13a, whereas Cas13b displays a distinct arrangement (Fig. 7a–e). The crRNA-target duplex is recognized by two HEPN and two Helical domains in Cas13d[31]; in Cas13a, it is recognized

inside the pocket formed by HEPN-1, HEPN-2, Helical-2, and Helical-3 domains[26]. As inferred from the charge distribution on the surface of protein, Cas13b may accommodate the crRNA-target duplex within the pocket formed by Helical-1, Helical-2, RRI-1, and HEPN-1 domains[38].

While Helical-1 domain of Cas13a and RRI-2 domain of Cas13b are responsible for pre-crRNA processing[25,38,39], their counterpart is lacking in the compact Cas13d. Instead, residues from HEPN-2 domain play a critical role in pre-crRNA processing by Cas13d. Therefore, HEPN-2 domain of Cas13d not only contributes a R-X$_4$-H motif for target RNA cleavage, but also provides a catalytic site for pre-crRNA cleavage.

The mature crRNAs show different recognition patterns within the Cas13a, Cas13b, and Cas13d binary complexes (Fig. 7a–h). In case of Cas13a, the repeat region of crRNA, which is characterized by stem-loop structure, is recognized by NTD and Helical-1 domains[25,26,39]. Without a counterpart of Helical-1 domain in Cas13a, the repeat region of crRNA is recognized by NTD and two HEPN domains in Cas13d, which still adopts a stem-loop structure. However, in Cas13d, about half of the stem-loop structure is exposed to the surrounding solvent, and the deletion of four Watson-Crick base pairs within this exposed region alters neither pre-crRNA processing nor target RNA cleavage. Notably, the nucleotides G(-9)-C(-1) of the repeat regions in both Cas13a and Cas13d exhibit a similar secondary structure and spatial U-shaped turn (Fig. 7f, g, i, j). A penta-hydrated Mg$^{2+}$ greatly stabilizes the U-shaped turn in the UrCas13d binary complex and possibly in the EsCas13d ternary complex, whereas a positively charged residue plays a similar role in stabilizing this region in the Cas13a ternary complex (Fig. 7i, j). Unlike the 5′-end repeat region of crRNA in the Cas13a and Cas13d complexes, the repeat region is located at the 3′-end of crRNA in the Cas13b binary complex and shows a more complicated architecture containing four sub-regions, which is recognized by Helical-2, two RRI domains and the linker region in Cas13b (Fig. 7c, h)[38]. Due to the fact that most of interactions between the crRNA repeat region and Cas13b exist in the recognition of the crRNA sugar-phosphate backbone, Cas13b is likely to recognize the crRNA repeat region in a structure-

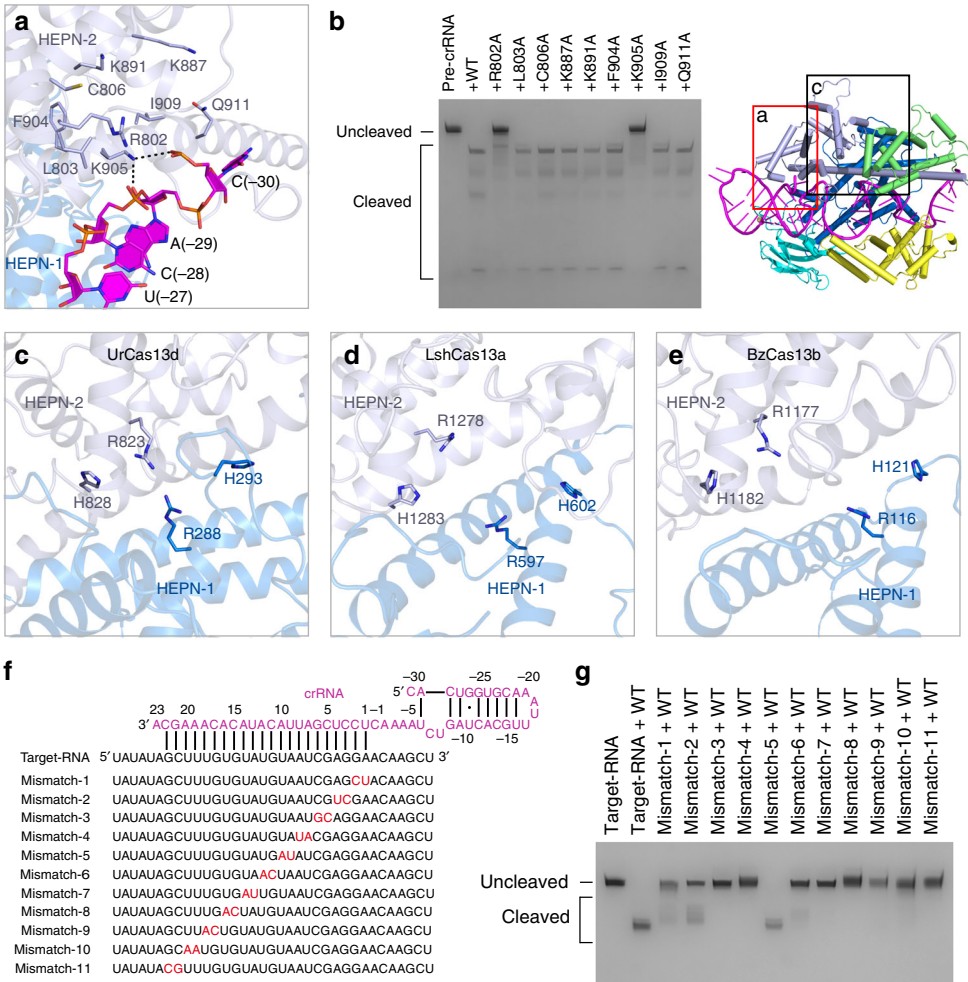

**Fig. 6** Active sites involved in pre-crRNA processing and target cleavage. **a** Close-up view of UrCas13d active site for pre-crRNA processing. Hydrogen bonds are shown as dashed lines. Domains and residues are colored according to Fig. 1. **b** Denaturing gel shows pre-crRNA cleavage by wild-type UrCas13d and the mutants speculatively involved in pre-crRNA processing. **c** Close-up view of UrCas13d active site for target RNA cleavage (PDB code: 6IV9). Residues A288 and A823 of the UrCas13d R288A/R823A binary complex was virtually mutated back to R288 and R823. **d** Close-up view of LshCas13a (*Leptotrichia shahii* Cas13a) active site for target RNA cleavage (PDB code: 5WTK). **e** Close-up view of BzCas13b (*Bergeyella zoohelcum* Cas13b) active site for target RNA cleavage. Residue A1177 of the BzCas13b R1177A binary complex (PDB code: 6AAY) was virtually mutated back to R1177. **f** Schematic representations of crRNA, target RNA and the mismatched targets designed for testing the mismatch tolerance between the crRNA spacer region and target RNA in UrCas13d. **g** Denaturing gel demonstrating the cleavage of target RNA and the mismatched target RNAs by wild-type UrCas13d in complex with crRNA

specific rather than sequence-specific manner[38]. By contrast, in Cas13a and Cas13d, both structure and sequence of the crRNA repeat region are specifically recognized and critical for target RNA cleavage.

The pre-crRNA processing by Cas13a proceeds in a divalent metal-ion-independent manner, whereas the divalent metal ion is indispensable for target RNA cleavage[17,18,39]. However, two different views on the essentiality of $Mg^{2+}$ for the pre-crRNA processing have been reported[21,31]. Here, we find that, similar to Cas13a, the divalent metal ion is not essential for pre-crRNA cleavage, but is critical for target cleavage by Cas13d. In the present study, two hydrated $Mg^{2+}$ ions were found to be important for stabilizing the conformation of the crRNA repeat region, out of which the $(Mg(H_2O)_5)^{2+}$ ion makes more important contributions. This suggests that the divalent metal ion may not be directly involved in catalyzing Cas13d-guided target RNA cleavage, but they could be important for maintaining the Cas13d-crRNA-target RNA complex in the conformation suitable for target cleavage.

In this study, we found that correct base pairings within the internal and the 3′-end spacer regions are essential for target recognition in Cas13d. The seed region may exist in either or both of these separate spacer regions in UrCas13d. In Cas13a, a central seed region has been determined[17,26,37]. Interestingly, a HEPN-nuclease switch region within the spacer has been identified adjacent to the seed region in Cas13a, and mismatches within this switch region may impede the conformational activation of Cas13a[37]. Whether the HEPN-nuclease switch region exists in Cas13d needs further investigation. In the Cas13b binary complex, given that the central region of the spacer is disordered and exposed to the surrounding solvent[38], it is possible that Cas13b uses this region of the spacer for target interrogation. Further studies are required to determine the exact seed region of Cas13b.

In summary, the present study determines the pre-crRNA processing site, explains the importance of two hydrated $Mg^{2+}$ ions for stabilizing the conformation of the crRNA repeat region, reveals the dependence of target RNA cleavage on structure and sequence of the crRNA repeat region, and identifies the mismatch

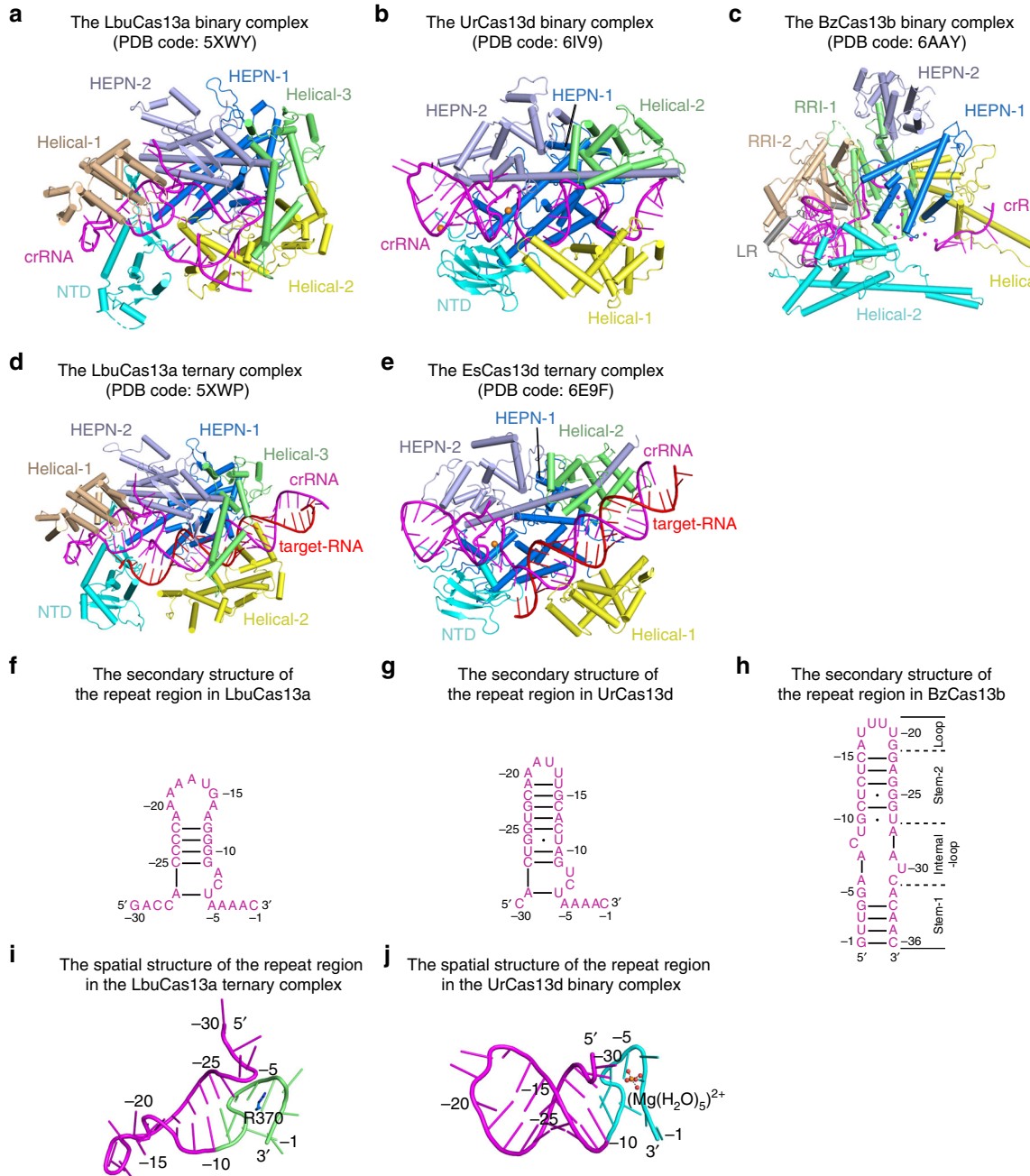

**Fig. 7** Comparison of Cas13a, Cas13b, and Cas13d. **a** Domain architectures of the LbuCas13a (*Leptotrichia buccalis* Cas13a) binary complex (PDB code: 5XWY). **b** Domain architectures of the UrCas13d binary complex (PDB code: 6IV9). **c** Domain architectures of the BzCas13b binary complex (PDB code: 6AAY). **d** Domain architectures of the LbuCas13a ternary complex (PDB code: 5XWP). **e** Domain architectures of the EsCas13d (*Eubacterium siraeum* Cas13d) ternary complex (PDB code: 6E9F). **f–h** Schematic representations of the secondary structures of the crRNA repeat regions in LbuCas13a, UrCas13d and BzCas13b, respectively. Bars between nucleotide pairs represent Watson-Crick base pairs and dots represent wobble G-U base pairs. **i** The structure of the crRNA repeat region in the LbuCas13a ternary complex. The nucleotides G(-9)-C(-1) in the repeat region are colored in green. Residue R370 of LbuCas13a is shown and labeled. **j** The structure of the crRNA repeat region in the UrCas13d binary complex. The nucleotides G(-9)-C(-1) in the repeat region are colored in cyan. The penta-hydrated $Mg^{2+}$ is shown and labeled

tolerance between the crRNA spacer region and target RNA. Our findings provide valuable information for rational engineering that would aid in developing the CRISPR-Cas13d system into an efficient tool for research, therapeutics and biotechnology.

## Methods

**Purification of the UrCas13d binary complex assembled in vivo.** The full-length gene encoding UrCas13d was synthesized (Supplementary Table 2) and inserted between *Nde*I and *Xho*I sites of the pET-30b vector. CRISPR template was synthesized (Supplementary Table 3) and cloned between *Nde*I and *Xho*I sites of the pCDFDuet-1 vector.

To acquire the binary complex assembled in vivo, vectors encoding the UrCas13d R288A/R823A mutant and CRISPR template were co-transformed into *E.coli* Rosetta (DE3) cells (Novagen). The cells were then grown at 37 °C for 3 h until the cell concentration reached $OD_{600}$ of approximately 0.6, which was followed by induction with 0.2 mM IPTG at 18 °C for 12 h. The cells were harvested and lysed in buffer A (25 mM Tris-HCl (pH 7.5), 500 mM NaCl, 5 mM β-mercaptoethanol, 1 mM phenylmethanesulfonyl fluoride (PMSF)). The lysate was centrifuged and the supernatant was loaded onto a Ni-NTA Superflow (QIAGEN) column. The nickel column was washed with buffer A containing 10

mM imidazole, after which the binary complex was eluted with buffer A containing 50 mM imidazole. Afterwards, the eluted target complex was successively purified with a Q FF column and a Heparin HP column (GE Healthcare) using a salt gradient between buffer B1 (25 mM Tris-HCl (pH 7.5), 150 mM NaCl, 2 mM MgCl$_2$, 2 mM dithiothreitol (DTT)) and buffer B2 (25 mM Tris-HCl (pH 7.5), 1 M NaCl, 2 mM MgCl$_2$, 2 mM DTT) in both steps. The target complex was then dialyzed against buffer C1 (25 mM Tris-HCl (pH 7.5), 200 mM NaCl, 2 mM MgCl$_2$, 2 mM DTT) and further purified using a Superdex 200 Increase (10/300 GL) column (GE Healthcare). Fractions containing the pure binary complex were collected, concentrated and aliquoted before storing at −80 °C for later crystallization uses. The SeMet-labeled binary complex was expressed in *E. coli* Rosetta (DE3) cells grown in M9 minimal medium supplemented with SeMet, Lys, Phe, Thr, Val, Leu, and Ile. The SeMet UrCas13d binary complex was purified following the procedure described above.

**Purification of the uncomplexed UrCas13d.** Full-length UrCas13d gene was expressed in *E. coli* Rosetta (DE3) cells (Novagen). After the cells were harvested and lysed in buffer A, the target protein was purified with a Ni-NTA Superflow (QIAGEN) column using buffer A and then with the SP FF column (GE Healthcare) using a salt gradient between buffer B3 (25 mM Tris-HCl (pH 7.5), 250 mM NaCl, 2 mM DTT) and buffer B4 (25 mM Tris-HCl (pH 7.5), 1 M NaCl, 2 mM DTT). Fractions of UrCas13d were dialyzed against buffer B3, concentrated, aliquoted and stored for later pre-crRNA or target RNA cleavage assays.

**In vitro synthesis and purification of RNA fragments.** T7 RNA polymerase was used in this study for in vitro synthesis of RNA fragments. Briefly, the synthetic DNA fragments encoding the sequences of pre-crRNA, crRNA, and target RNA were individually cloned into a modified pUC-119 vector between *Stu*I and *Hind*III sites[40]. After the recombinant vector was amplified in *E.coli* DH5α cells (Novagen), it was linearized by *Hind*III, and then purified using phenol-chloroform extraction and ethanol precipitation. Transcription reactions were performed at 37 °C for 3 h, in a buffer composed of 100 mM HEPES-K (pH 7.9), 20 mM MgCl$_2$, 30 mM DTT, 3 mM NTPs, 2 mM spermidine, 30 ng/μl linearized template, and 100 μg/ml home-made T7 RNA polymerase. The reaction sample was resolved in denaturing polyacrylamide gel containing 8 M urea. Target RNA bands were excised from the gel and eluted using the Elutrap system (GE Healthcare). The eluted RNA was collected by ethanol precipitation, dissolved in DEPC-treated H$_2$O and preserved at −80 °C.

**Site-directed mutagenesis.** In order to generate desired mutants (Supplementary Tables 3, 4, and 5), PCR reactions using mutation-specific oligonucleotides and vectors encoding the wild-type UrCas13d, pre-crRNA, crRNA, or target RNA as templates were carried out. The results of mutagenesis were verified by sequencing.

**Crystallization.** The SeMet-labeled UrCas13d R288A/R823A binary complex assembled in vivo was crystallized by the hanging-drop vapor diffusion method at 16 °C. Crystals were obtained by mixing 1 μl of complex solution (A$_{280\,nm}$ = 11.8) and 1 μl of reservoir solution (0.5% (w/v) Tryptone, 0.1 M HEPES sodium (pH 7.0), 16% (w/v) Polyethylene glycol 3350). Crystals grew to their full size within 6 days, after which they were harvested, cryoprotected in reservoir solution supplemented with 20% (v/v) glycerol and flash-frozen.

In addition, the unlabeled UrCas13d R288A/R823A binary complex assembled in vivo was crystallized by the hanging-drop vapor diffusion method at 16 °C. Crystals were obtained by mixing 1 μl of complex solution (A$_{280\,nm}$ = 11.7) and 1 μl of reservoir solution (8% (v/v) Tacsimate (pH 7.0), 17% (w/v) Polyethylene glycol 3350). Crystals grew to their full size within 7 days, after which they were harvested, cryoprotected in reservoir solution supplemented with 20% (v/v) glycerol and flash-frozen.

**Data collection and structure determination.** X-ray diffraction data was collected at Shanghai Synchrotron Radiation Facility (SSRF) on beamline BL-17U1. Datasets were processed automatically by the program XIA2 integrated into data collection platform of the beamline[41], and scaled by the program AIMLESS in CCP4[42,43]. The crystal structure of the SeMet-labeled UrCas13d R288A/R823A binary complex was solved by single-wavelength anomalous dispersion method with the program AutoSol in PHENIX[44]. An asymmetric unit of this SeMet-labeled crystal structure contains two binary complexes. Next, using the SeMet-labeled structure as a model, the crystal structure of the unlabeled UrCas13d R288A/R823A binary complex was solved by the molecular replacement method with the program Phaser-MR in PHENIX[44]. An asymmetric unit of this crystal structure contains one binary complex. Model building was carried out in COOT[45]. Multiple rounds of refinement were performed with REFMAC in CCP4[43,46]. All figures of the structures were prepared using PyMOL (http://pymol.org). Data collection and refinement statistics are listed (Supplementary Table 1).

**Pre-crRNA cleavage assay.** Pre-crRNA and the mutants were designed to contain two repeat regions separated by a spacer region (Supplementary Table 3). Reaction mixtures were prepared by incubating 9.6 μM purified UrCas13d with 4.8 μM pre-crRNA or the pre-crRNA mutants in the assay buffer (25 mM Tris-HCl (pH 7.5), 250 mM NaCl, 2 mM MgCl$_2$, 2 mM DTT). Total volume of the reaction mixture was adjusted to 20 μl. The cleavage was allowed to proceed at 37 °C for 30 min, after which the reactions were terminated in liquid nitrogen and subsequently quenched at 75 °C for 5 min by adding 2× loading buffer (2× TBE buffer, 12 M urea). Samples were then run on a 20% PAGE denaturing gel and the cleavage products were visualized with the toluidine blue staining. For pre-crRNA cleavage in the presence of different metal ions or ethylenediaminetetraacetic acid (EDTA), the assay buffer contained 2 mM divalent metal ions or 10 mM EDTA without 2 mM DTT. Each cleavage assay was independently carried out three times to verify the repeatability of the results.

**Target RNA cleavage assay.** Reaction mixtures were prepared by incubating 1.6 μM UrCas13d with 0.8 μM crRNA or the crRNA mutants in the assay buffer (25 mM Tris-HCl (pH 7.5), 250 mM NaCl, 2 mM MgCl$_2$, 2 mM DTT) at 37 °C for 15 min, and then a final concentration of 16 μM target RNA or the mismatched target RNAs was mixed with the prepared UrCas13d and crRNA sample at 20:2:1 molar ratio. Total volume of the reaction mixture was adjusted to 20 μl. The cleavage assays were then carried out at 37 °C for 45 min. For target RNA cleavage in the presence of different metal ions or EDTA, the assay buffer contained 2 mM divalent metal ions or 10 mM EDTA without 2 mM DTT. Each cleavage assay was independently carried out three times to verify the repeatability of the results. Reactions were terminated and samples were analyzed as described for the pre-crRNA cleavage assay.

**Reporting summary.** Further information on research design is available in the Nature Research Reporting Summary linked to this article.

## Data availability

The atomic coordinates and structure factors of the SeMet-labeled UrCas13d binary complex and the unlabeled UrCas13d binary complex have been deposited in the Protein Data Bank under the accession codes 6IV8 and 6IV9. The source data underlying Figs. 2c, f, g, 4c, e–h, 5e, f, 6b, g and Supplementary Figs. 6, 7a, b, e, f, 8 are provided as a Source Data file. Other data are available from the corresponding author upon reasonable request.

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

## Acknowledgements
This work was supported by the National Natural Science Foundation of China Grants (31770948, 31570875, and 31200559) and the Special Funds of the Central Government Guiding Local Science and Technology Development (2017L3009). We thank Fujian Normal University for financial support (Z0210509). The diffraction data were collected at the beamline BL-17U1 of Shanghai Synchrotron Radiation Facility (SSRF). We thank Xiao-Jian Hu and Huan Zhou for helpful discussions. We thank Zhijian J. Chen, Zhao-Qing Lou and Ning Gao for critical review of the manuscript.

## Author contributions
B.Z., Y.Y. and W.Y. purified the UrCas13b-crRNA complex and the UrCas13d mutants. B.Z. and Y.Y. purified the in vitro transcribed RNAs and performed the cleavage assays. B.Z., Y.Y. and W.Y. performed the crystallization screening. B.Z. optimized the crystallization condition, collected X-ray diffraction data and solved the crystal structure. H. J., Y.C., Y.L., J.C., J.L. and S.W. contributed to some experiments. B.Z., S.O., Q.C. and Y.S. H. designed the experiments. S.O. and B.Z. supervised the study. B.Z., V.P. and S.O. wrote the manuscript.
