## [Peer Review File · Nature Communications]

Reviewers' comments:

Reviewer #1 (Remarks to the Author):

Authors describe crystal structure and complementary biochemical analysis of *Ruminococcus* sp. Cas13d (UrCas13d). Although structural studies of a homologous Cas13d (Es) molecule were previously reported, the 1.8 Å resolution and the observed differences make this study novel. This study identifies important roles of divalent ions and protein residues in stabilizing the unique features in crRNA and the possible two seed regions within the guide RNA. The complementary biochemical analysis on the roles of RNA and protein elevates the structural findings. However, authors should address the following concerns:

- 1) The crystal structure contains 23 bases in the spacer region whereas the starting CRISPR RNA contains more than 23 bases. Have authors considered examining if the extra spacer RNA were removed by *E. coli* enzymes or they are simply disordered in the crystal structure?
- 2) Authors should point out that the stem-loop structure of the repeat is remarkably consistent with what mFold can predict with one exception for A-29.
- 3) The U-shaped turn for A(-2) – A(-5) is unique and important for UrCas13d function. In description of structure in this region, authors pointed out hydrogen bonding and polar interactions but did not mention if these adenosine nucleotides have unusual conformations with respect to their sugar pucker or glycosidic bond. In fact, authors did not mention if any other nucleotide has unusual conformations. Please clarify if there is simply lack of deviation from A-form RNA or negligence in description;
- 4) On page 6, authors should be clear on which groups of MgH₂O ion can form hydrogen bonds with the surrounding functional groups;
- 5) In the method section, it is unclear why pre-crRNA cleavage reaction was first terminated by adding liquid nitrogen and then heated to 75 degree. Please clarify;
- 6) Description of target RNA cleavage is also not clear. For instance, incubating 1.6 microM UrCas13d with 0.8 microM crRNA appears to make a 2:1 molar ratio of enzyme and crRNA. But later a molar ratio of 10:1:0.5 with was used, which seems contradicting to the 2:1 ratio of the pre-made RNP. Please clarify;
- 7) It is customary to spell out chemicals upon first mention (such as PMSF, DTT, SeMet etc.);
- 8) Figure 1, label the positions of the R-X4-H motifs. This helps with later discussions;
- 9) Why did authors describe Se-Met structure (for instance, Figure 2b) rather than the wild-type structure that is at a higher resolution?
- 10) In Fig 2c. the title should be "detailed base pair interactions at the U-shape turn region";
- 11) Supplementary Table 1, the Rmerge value for the highest resolution shell does not seem to be correct. It would be 167%.

Reviewer #2 (Remarks to the Author):

Zhang et al. report the crystal structure of Cas13d from uncultured *Ruminococcus* sp. (UrCas13d) in complex with a mature CRISPR RNA (crRNA). Cas13d is a family of compact type VI Cas endonucleases, and has significant potential for biotechnological tools targeting RNA. This manuscript reports the highest resolution structure of the Cas13d-crRNA binary structure available to date. The structure reveals many interactions between the RNA, protein, and two structurally important, hydrated magnesium ions. The authors test an extensive set of mutants in both Cas13d

and the crRNA to determine which interactions are important. They also identify the putative catalytic residues involved in pre-crRNA biogenesis and show that this activity is metal-independent. Finally, they test a set of target mutations to determine whether a "seed" region is present in the crRNA.

This study follows a previous report of the cryo-EM structures of the *Eubacterium siraeum* apo, binary and ternary Cas13d complexes (reference 31 in the manuscript). The structure is reported at higher resolution than the binary EsCas13d complex, although it is a bit unclear whether this improved resolution provides much additional data outside of the hydration state of the magnesium ions. The mutagenesis data probing the interactions between the crRNA and protein is more extensive than in any prior study, although the presentation is at times difficult to parse. The evidence for two seed regions for UrCas13d is a bit underdeveloped and could use further explanation. These issues, along with other major and minor concerns, are detailed below and should be addressed by the authors.

Organization of the manuscript:

As mentioned above, the presentation of the RNA-protein interactions and mutagenesis data is very extensive, and the authors tested an impressive number of Cas13d and crRNA mutants. The sheer number of mutants can make the paper a bit hard to follow at points. The authors could alleviate this issue to some degree by reorganizing the Results section to present the data in a more cohesive manner. For example, the first three sentences of Section 5 of the Results section reiterate points that were already made in Sections 2 and 3. These three sections could be reorganized into one single section describing the structure of this region of the crRNA repeat and how this structure is stabilized by the Mg²⁺ ions and protein interactions.

Similarly, the authors could reorganize figures to help with the flow of the manuscript. The authors reference some figure panels out of order (e.g. Figure 3h is referenced before Figures 3c-e), and some figure panels are barely referenced and never described in the text (e.g. Fig. 2d-f) or redundant (e.g. Fig. 1b and 2a). Re-ordering or removing figure panels could help the reader to follow the text better.

Major concerns about data:

1. The authors do not draw many conclusions from the mutagenesis data, nor do they convincingly explain why having such detailed information on which interactions stabilize the Cas13d-crRNA complex is important. In fact, there are some interesting results that go undiscussed. For example, the authors show that some Cas13d residues (D178, K181 and K524) that interact with the -2 to -8 region of the crRNA are essential for target cleavage (Figure 3h) but not for pre-crRNA cleavage (Supplementary Figure 5). Similarly, Mg²⁺ is required for target cleavage but not crRNA maturation. This suggests that the structure formed in this region of the repeat may have mechanistic consequences for target RNA binding or cleavage, but not for pre-crRNA binding or cleavage. Have the authors tested whether these mutants remain bound to the crRNA following pre-crRNA cleavage? If so, have they tested whether Cas13d-crRNA can bind to target RNA in the absence of Mg²⁺ or in the presence of the destabilizing mutants?

2. The putative seed sequence data is not convincing. A seed sequence should be defined as a region of the crRNA that nucleates binding to the target. In order to truly define the seed, an experimental strategy that takes binding into account (e.g. binding or competition assay) must be used. However, the authors only test the effects of target mutations on cleavage. It is possible that mismatches between crRNA and target ablate cleavage through a mechanism other than blocking target binding. Therefore, while these data may reveal the mismatch tolerance (or lack thereof) at certain positions of the crRNA, it does not reveal whether a seed is present for UrCas13d. The authors could perform additional experiments to determine the seed, or they could rewrite their interpretation of this data and remove mention of a seed.

3. The authors show that part of the crRNA stem-loop can be deleted. This potentially has some use for engineering crRNAs with shorter or longer stem-loops for applications. However, the authors only test this truncation at a single time point. It would be more appropriate to test the

rate of pre-crRNA biogenesis and target RNA cleavage for this mutant versus the full-length crRNA to determine whether or not this truncation has an effect on Cas13d activity.

Minor concerns:

1. What are the relative sizes of Ur and EsCas13d? The authors mention that UrCas13d “exhibits an atypically compact architecture (lines 24-25, p. 4).” However, the architecture does not appear to be significantly different from EsCas13d based on Supplemental Figure 7. This section could be revised, and information on the relative lengths of the two proteins could be added.

2. For the pre-crRNA cleavage and target RNA cleavage assays shown in Figure 3, what do the products correspond to? The authors could provide a diagram of the expected product sizes for the pre-crRNA. What is the length of the product of target RNA cleavage? The product appears to be discrete, rather than the non-specific cleavage pattern that is typically observed for Cas13 proteins. Is the main product a fully degraded RNA?

3. In a previous study (reference 21 of the manuscript), UrCas13d was abbreviated as RspCas13d. The authors could mention this alternate nomenclature when first abbreviating UrCas13d (e.g. “UrCas13d, also known as RspCas13d”).

4. On p. 3, line 14-15, the authors mention Cas13 as the only family in Class 2 that targets RNA. Some Cas9 variants have also been shown to target RNA (see PMID: 29303478, 29456189, 29499139). The authors could mention these Cas9 variants, or change the text to say that Cas13 is the only family that exclusively targets RNA.

5. In figures and legends, the authors refer to the (U(-24)-A(-21)/U(-16)-A(-13)) deletion mutant as the Δ stem mutant. The authors could also mention this nomenclature in the text (e.g. p. 7, line 1).

6. Reference 31 is out of order.

7. On p. 5, the authors refer to the “NTD domain” twice. “Domain” is redundant with the D in “NTD” and can be removed.

Point-by-point response to Reviewers' comments

The authors thank the reviewers for professional and insightful review. Following the reviewer's suggestions, the authors have reorganized the manuscript. The former sections 2, 3 and 5 have been rearranged into new sections 2 and 3. These rearranged sections are highlighted in deep orange color. Changes in the revised manuscript are highlighted in blue color. For clarity, the former Figures 1-3 have been reorganized. Also, the former Figure 4d and Supplementary Fig. 2 have been regenerated to be consistent with the revised contents of the manuscript. Other changes in the manuscript and figures have been described in the following point-by-point responses.

Reviewer #1 (Remarks to the Author):

Authors describe crystal structure and complementary biochemical analysis of *Ruminococcus* sp. Cas13d (UrCas13d). Although structural studies of a homologous Cas13d (Es) molecule were previously reported, the 1.8 Å resolution and the observed differences make this study novel. This study identifies important roles of divalent ions and protein residues in stabilizing the unique features in crRNA and the possible two seed regions within the guide RNA. The complementary biochemical analysis on the roles of RNA and protein elevates the structural findings.

However, authors should address the following concerns:

Point-by-point responses to Reviewer-1's comments:

1) The crystal structure contains 23 bases in the spacer region whereas the starting CRISPR RNA contains more than 23 bases. Have authors considered examining if the extra spacer RNA were removed by *E. coli* enzymes or they are simply disordered in the crystal structure?

Authors' response:

In response to the reviewer's comment, the authors have extracted RNA samples from the purified UrCas13d binary complex assembled *in vivo* by using phenol-chloroform extraction. The extracted RNA samples were sent to GenePharma Company (Suzhou, China) for the mass-spectrometry analysis. The mass-spectrometry result shows that the measured values of two major peaks approximately correspond to molecular weights of 52-nt and 53-nt mature crRNAs (see figure below).

In this manuscript, the authors reported two binary complexes. In the SeMet-labeled binary complex, a mature 53-nt crRNA was observed, which implied that the extra spacer RNA was removed by *E. coli* enzyme(s). In the native binary complex, a mature 50-nt crRNA was observed, which suggested that the last 2-nt or 3-nt of spacer RNA were simply disordered in the crystal structure.

The calculations of molecular weights of mature crRNAs and the mass-spectrometry result is presented below:

53-nt crRNA sequence includes 30-nt repeat region (underlined) and 23-nt spacer region:

CACUGGUGCAAUUUGCACUAGUCUAAAACUCCUCGAUUACAUACACAAAGCA

Theoretical 53-nt RNA Molecular Weight (MW):

$$= (A_n \times 329.21) + (U_n \times 306.17) + (C_n \times 305.18) + (G_n \times 345.21)$$

$$= (19 \times 329.21) + (13 \times 306.17) + (14 \times 305.18) + (7 \times 345.21)$$

$$=6254.99+3980.21+4272.52+2416.47$$

$$=16924.19 \text{ (g/mol)}$$

52-nt crRNA sequence includes 30-nt repeat region (underlined) and 22-nt spacer region:
CACUGGUGCAAUUUGCACUAGUCUAAAACUCCUCGAUUACAUAACACAAAGC

Theoretical 52-nt RNA Molecular Weight (MW):

$$= (A_n \times 329.21) + (U_n \times 306.17) + (C_n \times 305.18) + (G_n \times 345.21)$$

$$= (18 \times 329.21) + (13 \times 306.17) + (14 \times 305.18) + (7 \times 345.21)$$

$$=6254.99+3980.21+4272.52+2416.47$$

$$=16594.98 \text{ (g/mol)}$$

2) Authors should point out that the stem-loop structure of the repeat is remarkably consistent with what mFold can predict with one exception for A-29.

Authors' response:

Following the reviewer's suggestion, the authors have added the Supplementary Fig. 3 and one sentence stating that "Its secondary structure is remarkably similar to the predicted one, with the exception of base pairing of A(-29) (Fig. 1b and Supplementary Fig. 3)." in the revised manuscript.

3) The U-shaped turn for A(-2) – A(-5) is unique and important for UrCas13d function. In description of structure in this region, authors pointed out hydrogen bonding and polar interactions but did not mention if these adenosine nucleotides have unusual conformations with respect to their sugar pucker or glycosidic bond. In fact, authors did not mention if any other nucleotide has unusual conformations. Please clarify if there is simply lack of deviation from A-form RNA or negligence in description;

Authors' response:

The authors have neglected to describe the unusual conformations of the nucleotides within the stem-loop structure of crRNA. The authors thank the reviewer for reminding us about this point.

For clarity, “The stem region shows a distorted RNA duplex” was revised to “The stem region shows a distorted A-form RNA duplex” in the revised manuscript.

Also, the authors have reinspected the crRNA structure and tried to find unusual conformations of the nucleotides, especially within the U-shaped turn region of U(-8)-A(-2). Indeed, compared to the A-form RNA, the nucleotides U(-8), A(-4) and A(-3) possess unusual conformations. Therefore, the authors have added the following sentence to the revised manuscript: “It should be noted that, when compared to the standard A-form RNA, nucleotides U(-8), A(-4) and A(-3) within the U-shaped turn have unusual conformations with respect to their sugar pucker or glycosyl bond (Supplementary Fig. 5).”

4) On page 6, authors should be clear on which groups of $Mg(H_2O)_5^{2+}$ ion can form hydrogen bonds with the surrounding functional groups;

Authors' response:

The detailed interactions of the key water molecules within the $Mg(H_2O)_5^{2+}$ ion were labeled in Fig. 3 of the revised manuscript. The authors have also added one sentence in the revised manuscript which states the following: “The detailed interactions between the $Mg(H_2O)_5^{2+}$ ion and the surrounding chemical groups are shown in Fig. 3”.

5) In the method section, it is unclear why pre-crRNA cleavage reaction was first terminated by adding liquid nitrogen and then heated to 75 degree. Please clarify;

Authors' response:

The cleavage reactions can be stopped synchronously by using liquid nitrogen and the reaction time can be controlled more accurately. In practice, it takes some time to add 2× loading buffer to each reaction sample before quenching the reaction at 75°C. If the reaction samples were first flash frozen in liquid nitrogen, the authors will not worry about different time delays prior to quenching the reaction at 75°C, thus improving the repeatability of our assay results.

6) Description of target RNA cleavage is also not clear. For instance, incubating 1.6 microM UrCas13d with 0.8 microM crRNA appears to make a 2:1 molar ratio of enzyme and crRNA. But later a molar ratio of 10:1:0.5 with was used, which seems contradicting to the 2:1 ratio of the pre-made RNP. Please clarify;

Authors' response:

The authors feel sorry for not expressing the meaning clearly and for causing the misunderstanding. A molar ratio of 10:1:0.5 (target RNA:protein:crRNA) seems contradicting to the molar ratio 2:1 (protein:crRNA), but actually the molar ratio of 10:1:0.5 is equal to 20:2:1. For clarity, the authors have revised “10:1:0.5 molar ratio” to “20:2:1 molar ratio” in the revised manuscript.

7) It is customary to spell out chemicals upon first mention (such as PMSF, DTT, SeMet etc.);

Authors' response:

Following the reviewer's suggestion, the authors have spelled out PMSF, DTT, SeMet, EDTA and RMSD on their first appearances in the revised manuscript.

8) Figure 1, label the positions of the R-X4-H motifs. This helps with later discussions;

Authors' response:

Following the reviewer's suggestion, the authors have labeled two R-X₄-H motifs in Fig. 1a.

9) Why did authors describe Se-Met structure (for instance, Figure 2b) rather than the wild-type structure that is at a higher resolution?

Authors' response:

As mentioned in the response to the reviewer's comment (1), the SeMet-labeled binary complex contains a 53-nt mature crRNA, and the high-resolution native binary complex contains a 50-nt mature crRNA. In the previous manuscript, when the authors described the information related to the 53-nt crRNA, the SeMet-labeled structure was used. To avoid the misunderstanding and follow the suggestion of the other reviewer, the authors merged Fig. 1 and Fig. 2, and focused on describing the high-resolution native structure in the revised manuscript.

10) In Fig 2c. the title should be “detailed base pair interactions at the U-shape turn region”;
Authors response:

Authors' response:

In response to other reviewer's comment, the authors have reorganized the manuscript and Figures 1-3. For better organization, the former Fig. 2c has been deleted in the revised manuscript.

11) Supplementary Table 1, the R_{merge} value for the highest resolution shell does not seem to be correct. It would be 167%.

Authors' response:

The X-ray diffraction dataset was automatically processed by the program Xia2 integrated into data collection platform of the beamline BL-17U1 at Shanghai Synchrotron Radiation Facility (SSRF). The R_{merge} value was collected from the data-processing result of the program Xia2. To decrease the R_{merge} value for the highest resolution shell, the authors rescaled the unmerged dataset using the program AIMLESS in CCP4, and a little decrease in the highest resolution of the

dataset can significantly decrease the R_{merge} value for the highest resolution shell (e.g. from 167% to 84.2%). Besides, this rescaling does not perceptibly alter the fitness between the PDB model and the density map. The authors also generated an updated Supplementary Table S1 in which data collection and refinement statistics are described.

Reviewer #2 (Remarks to the Author):

Zhang et al. report the crystal structure of Cas13d from uncultured *Ruminococcus* sp. (UrCas13d) in complex with a mature CRISPR RNA (crRNA). Cas13d is a family of compact type VI Cas endonucleases, and has significant potential for biotechnological tools targeting RNA. This manuscript reports the highest resolution structure of the Cas13d-crRNA binary structure available to date. The structure reveals many interactions between the RNA, protein, and two structurally important, hydrated magnesium ions. The authors test an extensive set of mutants in both Cas13d and the crRNA to determine which interactions are important. They also identify the putative catalytic residues involved in pre-crRNA biogenesis and show that this activity is metal-independent. Finally, they test a set of target mutations to determine whether a “seed” region is present in the crRNA.

This study follows a previous report of the cryo-EM structures of the *Eubacterium siraeum* apo, binary and ternary Cas13d complexes (reference 31 in the manuscript). The structure is reported at higher resolution than the binary EsCas13d complex, although it is a bit unclear whether this improved resolution provides much additional data outside of the hydration state of the magnesium ions. The mutagenesis data probing the interactions between the crRNA and protein is more extensive than in any prior study, although the presentation is at times difficult to parse. The evidence for two seed regions for UrCas13d is a bit underdeveloped and could use further explanation. These issues, along with other major and minor concerns, are detailed below and should be addressed by the authors.

Organization of the manuscript:

As mentioned above, the presentation of the RNA-protein interactions and mutagenesis data is very extensive, and the authors tested an impressive number of Cas13d and crRNA mutants. The sheer number of mutants can make the paper a bit hard to follow at points. The authors could alleviate this issue to some degree by reorganizing the Results section to present the data in a more cohesive manner. For example, the first three sentences of Section 5 of the Results section reiterate points that were already made in Sections 2 and 3. These three sections could be reorganized into one single section describing the structure of this region of the crRNA repeat and how this structure is stabilized by the Mg^{2+} ions and protein interactions.

Similarly, the authors could reorganize figures to help with the flow of the manuscript. The authors reference some figure panels out of order (e.g. Figure 3h is referenced before Figures 3c-e), and some figure panels are barely referenced and never described in the text (e.g. Fig. 2d-f) or redundant (e.g. Fig. 1b and 2a). Re-ordering or removing figure panels could help the reader to follow the text better.

Authors' response:

Following the reviewer's suggestion, the authors have reorganized the manuscript. Sections 2, 3 and 5 have been rearranged into two sections, and we attempted to avoid repeated descriptions in the revised sections. Also, for clarity, the authors have reorganized figures. The previous Figures 1 and 2 have been simplified and combined into the new Figure 1. In addition, the previous Figure 3 has been divided into the new Figures 2 and 4. The new Figure 2 is focused on the structural and functional information related to the hydrated Mg^{2+} ions, and the new Fig. 4 includes the remainder of the previous Fig. 3. Moreover, in the revised manuscript, to ensure consistent order of the sample loading in the denaturing gel, the authors prepared new denaturing gel to demonstrate the results. Residues K51, F531, K799 and K377 are not present in the former structural figures and do not have significant impact on the target RNA cleavage. Therefore, in the revised manuscript, the cleavage results related to mutations K51A, F531A, K799A and K377A have been omitted from the corresponding figures.

Major concerns about data:

1. The authors do not draw many conclusions from the mutagenesis data, nor do they convincingly explain why having such detailed information on which interactions stabilize the Cas13d-crRNA complex is important. In fact, there are some interesting results that go undiscussed. For example, the authors show that some Cas13d residues (D178, K181 and K524) that interact with the -2 to -8 region of the crRNA are essential for target cleavage (Figure 3h) but not for pre-crRNA cleavage (Supplementary Figure 5). Similarly, Mg^{2+} is required for target cleavage but not crRNA maturation. This suggests that the structure formed in this region of the repeat may have mechanistic consequences for target RNA binding or cleavage, but not for pre-crRNA binding or cleavage. Have the authors tested whether these mutants remain bound to the crRNA following pre-crRNA cleavage? If so, have they tested whether Cas13d-crRNA can bind to target RNA in the absence of Mg^{2+} or in the presence of the destabilizing mutants?

Authors' response: To answer these questions, the authors performed D178A, K181A and K524A mutations in the dead UrCas13d (designated dUrCas13d), which contains the R288A/R823A mutation in its HEPN domains). The plasmids encoding dUrCas13d or the dUrCas13d D178A, K181A, K524A mutants were individually co-transformed with the vector encoding the CRISPR template into *E. coli* Rosetta(DE3) cells. dUrCas13d binary complex or the mutant binary complexes assembled *in vivo* were purified following the protocol described in the Methods. The gel filtration results (using the Superdex 200 increase (10/300 GL) column) show that these binary complexes have similar peak positions and A_{260}/A_{280} values, indicating that D178A, K181A or K524A mutations do not disturb mature crRNA binding following the pre-crRNA processing. The bound mature crRNAs within four binary complexes were further validated by denaturing gel electrophoresis. Furthermore, the authors performed *in vitro* pre-crRNA cleavage assays by using the protocol described in the Methods, and analyzed the results by native PAGE (6% native PAGE gel and 0.5xTBE buffer were used for electrophoresis). Pre-crRNA used in the cleavage assays contains two repeat regions separated by a spacer region. As shown in the figure below, following the pre-crRNA cleavage, the dUrCas13d D178A, K181A, K524A mutants bind mature crRNA with capacity similar to dUrCas13d.

Figure 1 (a to d) the gel-filtration purifications of the dUrCas13d-crRNA, the dUrCas13d-D178A-crRNA, the dUrCas13d-K181A-crRNA, and the dUrCas13d-K524A-crRNA binary complexes using the Superdex 200 increase (10/300 GL) column; the UV peak position of each binary complex was marked; (e) the SDS-PAGE result demonstrating the proteins within the dUrCas13d binary complex and the mutant binary complexes; values of A₂₈₀ and A₂₆₀/A₂₈₀ of each concentrated binary complex are shown; (f) the denaturing-gel result demonstrating the mature crRNAs within the dUrCas13d binary complex and the mutant binary complexes; (g) the native PAGE result shows that dUrCas13d and its D178A, K181A, K524A mutants have the similar binding capacity for mature crRNA following the pre-crRNA cleavage.

To test the capacity with which the dUrCas13d-crRNA complex in the presence or absence of Mg^{2+} binds target RNA, as well as capacities with which the mutant-crRNA complexes bind target RNA, the authors incubated the binary complexes with the target RNA at 22°C for 30 minutes in the buffer system (25 mM Tris-HCl (pH 7.5), 200 mM NaCl, 2 mM DTT) with 2 mM $MgCl_2$ for the dUrCas13d and the mutants binary complexes, or with 2mM EDTA for the dUrCas13d binary complex in the absence of Mg^{2+} . The target RNA concentration used was 6.2 μM , and three concentrations of each binary complex used were $A_{280}=0.25, 1.0$ and 4.0. EMSAs were then carried out to verify the binding capacities for respective complexes (6% native PAGE gel and 0.5xTBE buffer were used for electrophoresis). The results indicate that as the amount of the binary complex increases, the amount of the unbound target RNA decreases. The mutant binary complexes (D178A, K181A, K524A) or the dUrCas13d-crRNA binary complex in the absence of Mg^{2+} show target RNA binding capacities comparable to the dUrCas13d-crRNA complex.

EMSA Results:

Figure 1 | the EMSA results. (a) as the amount of the dUrCas13d or its mutant binary complexes increases, the amount of the unbound target RNA decreases; (b) the dUrCas13d-crRNA binary complex shows comparable binding capacity for target RNA in the presence and absence of Mg^{2+} .

Together with the relevant result described in the manuscript, the results above indicate that the absence of Mg^{2+} ions and mutations of D178, K181 or K524 to alanine do not significantly disturb the pre-crRNA processing and target RNA binding. Although the Mg^{2+} ion and residues D178, K181 and K524 play important roles in stabilizing the conformation of the U-shaped turn within the crRNA repeat region and are also essential for the target RNA cleavage, the authors can not rule out other possible mechanisms by which Cas13d utilizes the Mg^{2+} ion to accomplish the target RNA cleavage. In Cas13a, the Mg^{2+} ions are essential for the target RNA cleavage, but are not required for the pre-crRNA processing. In the Cas13a structure, no hydrated Mg^{2+} ion comparable to those in Cas13d was found to stabilize the crRNA repeat region, yet Cas13a still needs metal ions to carry out the target RNA cleavage.

2. The putative seed sequence data is not convincing. A seed sequence should be defined as a region of the crRNA that nucleates binding to the target. In order to truly define the seed, an experimental strategy that takes binding into account (e.g. binding or competition assay) must be used. However, the authors only test the effects of target mutations on cleavage. It is possible that mismatches between crRNA and target ablate cleavage through a mechanism other than blocking target binding. Therefore, while these data may reveal the mismatch tolerance (or lack thereof) at certain positions of the crRNA, it does not reveal whether a seed is present for UrCas13d. The authors could perform additional experiments to determine the seed, or they could rewrite their interpretation of this data and remove mention of a seed.

Authors' response:

The authors thank the reviewer's professional comment. At present, the authors have not developed an effective protocol to determine the seed region. Therefore, following the reviewer's second suggestion, the authors rewrote the interpretation of this data related to the seed region in the revised manuscript.

3. The authors show that part of the crRNA stem-loop can be deleted. This potentially has some use for engineering crRNAs with shorter or longer stem-loops for applications. However, the authors only test this truncation at a single time point. It would be more appropriate to test the rate of pre-crRNA biogenesis and target RNA cleavage for this mutant versus the full-length crRNA to determine whether or not this truncation has an effect on Cas13d activity.

Authors' response:

Following the reviewer's suggestion, the authors carried out the time-course experiments to explore the possible differences of the Cas13d activity on the pre-crRNA processing and target RNA cleavage for the wild-type pre-crRNA or crRNA versus their Δ stem mutants. The experimental results show that, compared to pre-crRNA or crRNA, the Δ stem mutants have a similar efficiency in the pre-crRNA processing and target RNA cleavage activities of Cas13d.

For the pre-crRNA processing, the authors chose time points of 1, 3, 5, 10, 20 and 30 minutes and the assays were carried out three times independently. The expected sizes of the pre-crRNA processing products were labeled. Besides, the authors tested the time-course pre-crRNA processing by the Cas13d K905A mutant in the presence of pre-crRNA or the Δ stem mutant as a control. The results show that the K905A mutant has almost lost its ability to process both pre-crRNA and the Δ stem mutant.

For the target RNA cleavage, the authors chose time points of 1, 3, 5, 10, 30 and 45 minutes and the assays were carried out three times independently. Surprisingly, both crRNA and its Δ stem mutant were highly efficient in guiding the Cas13d activity on the target RNA cleavage. Besides, the authors tested the time-course target RNA cleavage by the Cas13d D178A mutant in the presence of crRNA or the Δ stem mutant as a control. Both complexes exhibit significantly decreased target RNA cleavage activities.

Figure 1 | (a and b) denaturing gel demonstrating the time-course cleavage of pre-crRNA or the Δ stem mutant by the wild-type UrCas13d or the K905A mutant; (c and d) the expected sizes of the pre-crRNA cleavage products were labeled as i-vi, and the expected sizes of the Δ stem mutant cleavage products were labeled as vii-xi; (e and f) denaturing gel demonstrating the time-course cleavage of target RNA by the wild-type UrCas13d or the D178A mutant in the presence of crRNA or the Δ stem mutant. The rationale behind selecting these mutants in the experiments is that the K905A mutant abolishes the pre-crRNA processing, while the D178A mutant greatly decreases the target RNA cleavage.

Minor concerns:

1. What are the relative sizes of Ur and EsCas13d? The authors mention that UrCas13d “exhibits an atypically compact architecture (lines 24-25, p. 4).” However, the architecture does not appear to be significantly different from EsCas13d based on Supplemental Figure 7. This section could be revised, and information on the relative lengths of the two proteins could be added.

Authors’ response:

Following the reviewer’s suggestion, the authors have added the following sentence as following in the revised manuscript: “The overall architectures of both binary complexes are compact and similar, with the root-mean-square deviation (RMSD) of 1.62 Å between 735 aligned C α atoms of UrCas13d (922 amino acid residues) and EsCas13d (954 amino acid residues) (Supplementary Fig. 9a).”

2. For the pre-crRNA cleavage and target RNA cleavage assays shown in Figure 3, what do the products correspond to? The authors could provide a diagram of the expected product sizes for the pre-crRNA. What is the length of the product of target RNA cleavage? The product appears to be discrete, rather than the non-specific cleavage pattern that is typically observed for Cas13 proteins. Is the main product a fully degraded RNA?

Authors’ response:

Following the reviewer’s suggestion, the authors have provided the expected sizes of the pre-crRNA processing products in the Figure of our response to the reviewer’s “Major concern (3)” or in Supplementary Fig. 7 of the revised manuscript.

To determine the product size of target RNA cleavage, the authors have extracted products of target RNA cleavage from the cleavage assays by using phenol-chloroform extraction. The extracted RNA samples were sent to GenePharma Company (Suzhou, China) for the mass-spectrometry analysis. The mass-spectrometry result as the following figure shows that the measured values of two major peaks approximately correspond to molecular weights of ~16-nt and ~18-nt products.

To investigate whether the main product was a fully degraded RNA, the reaction time for the target cleavage was prolonged from 45 minutes to 90 minutes. The cleavage results show that even when the reaction time is doubled, the cleavage pattern remains unchanged (shown in the figure below), meaning that the main product has already been fully processed at the earlier time point.

3. In a previous study (reference 21 of the manuscript), UrCas13d was abbreviated as RspCas13d. The authors could mention this alternate nomenclature when first abbreviating UrCas13d (e.g. “UrCas13d, also known as RspCas13d”).

Authors’ response:

Following the reviewer’s suggestion, the authors have added “UrCas13d, also known as RspCas13d” to the revised manuscript.

4. On p. 3, line 14-15, the authors mention Cas13 as the only family in Class 2 that targets RNA. Some Cas9 variants have also been shown to target RNA (see PMID: 29303478, 29456189, 29499139). The authors could mention these Cas9 variants, or change the text to say that Cas13 is the only family that exclusively targets RNA.

Authors’ response:

The authors thank the reviewer for reminding us about this important feature of Cas9 variants. Following the reviewer’s suggestion, the authors have revised the manuscript as “Cas13 family is the only family of class 2 Cas enzymes known to exclusively target single-stranded RNA”.

5. In figures and legends, the authors refer to the (U(-24)-A(-21)/U(-16)-A(-13)) deletion mutant as the Δ stem mutant. The authors could also mention this nomenclature in the text (e.g. p. 7, line 1).

Authors’ response:

Following the reviewer’s suggestion, the (U(-24)-A(-21)/U(-16)-A(-13)) deletion mutant has been termed the Δ stem mutant in the revised manuscript.

6. Reference 31 is out of order.

Authors’ response:

Following the reviewer’s suggestion, the authors have revised the order of reference 31.

7. On p. 5, the authors refer to the “NTD domain” twice. “Domain” is redundant with the D in “NTD” and can be removed.

Authors’ response:

Following the reviewer’s suggestion, the authors have removed “Domain” after “NTD” in the revised manuscript.

REVIEWERS' COMMENTS:

Reviewer #1 (Remarks to the Author):

Authors addressed the concerns satisfactorily in the revised manuscript.

Reviewer #2 (Remarks to the Author):

Zhang et al. report the first <math><2\text{\AA}</math> resolution structure of Cas13d, a compact RNA-guided RNA targeting Cas endonuclease. The structure reveals several details about the RNA structure, its stabilization by hydrated Mg^{2+} , and the active sites of the protein. Following revision, the manuscript is easy to follow and the data are presented clearly. The authors have addressed all of my major concerns. I believe this work will be of broad interest to the readers of Nature Communications, and that it is suitable for publication.

Response to reviewers' comments:

REVIEWERS' COMMENTS:

Reviewer #1 (Remarks to the Author):

Authors addressed the concerns satisfactorily in the revised manuscript.

The authors thank the reviewer once more for all comments and suggestions, which notably improved the quality of our manuscript.

Reviewer #2 (Remarks to the Author):

Zhang et al. report the first $<2\text{\AA}$ resolution structure of Cas13d, a compact RNA-guided RNA targeting Cas endonuclease. The structure reveals several details about the RNA structure, its stabilization by hydrated Mg^{2+} , and the active sites of the protein. Following revision, the manuscript is easy to follow and the data are presented clearly. The authors have addressed all of my major concerns. I believe this work will be of broad interest to the readers of Nature Communications, and that it is suitable for publication.

The authors thank the reviewer for professional review of our manuscript and are happy to receive positive evaluation following revision based on the reviewer's concerns. We hope that your suggestions will make the manuscript and data easier to follow to all readers.